# The impact of anodization modification on titanium interaction with human osteoblasts and fibroblasts (in vitro study)

Kamonmard Moungthong[1], Daneeya Na Nan[2], Pravej Serichetaphongse[3,4], Wareeratn Chengprapakorn®[5]*

1 Department of Prosthodontics, Faculty of Dentistry, Chulalongkorn University, 34 Henri-Dunant Road, Wangmai, Patumwan, Bangkok, Thailand, 2 Division of Research Affairs and Center of Excellence for Dental Stem Cell Biology, Faculty of Dentistry, Chulalongkorn University, Bangkok, Thailand, 3 Faculty of Dentistry, Siam University, 38 Petchkasem Road, Bangwa, Phasicharoen, Bangkok, Thailand, 4 Dental Center, Bumrungrad International Hospital, Bangkok, Thailand, 5 Department of Prosthodontics, Faculty of Dentistry, Chulalongkorn University, 34 Henri-Dunant Road, Wangmai, Patumwan, Bangkok, Thailand

* wchengprapakorn@gmail.com

## Abstract

### Title

The impact of new anodization modification on titanium interaction with human osteoblasts and fibroblasts (in vitro)

### Purpose

To compare the behavior of osteoblast and fibroblast cells, as well as the surface characteristics, of modified anodized titanium surfaces at different voltages with sandblasted and acid-etched titanium surfaces.

### Methods

Twenty titanium discs were fabricated and subjected to 4 types of surface treatments: machined surface, sandblasted and acid-etched surface (SLA), modified anodized A surface at 30V DC and modified anodized B surface at 70V DC. Surface roughness and contact angle were measured. Osteoblast cells were seeded onto titanium discs and evaluated for cell proliferation, Alkaline phosphatase (ALP) activity, Alizarin red staining, and cell morphology with SEM imaging. Immunofluorescent staining was performed to assessed cell adhesion of both osteoblast and fibroblast cells. Statistic analysis was conducted using ANOVA with Tukey post hoc multiple comparisons.

### Results

The topography and osteoblast cell spreading on both modified anodized titanium surfaces differed significantly from machined and SLA surfaces ($p < 0.05$). However,

**Data availability statement:** All relevant data are within the paper and its Supporting Information files.

**Funding:** The author(s) received no specific funding for this work.

**Competing interests:** The authors have declared that no competing interests exist.

there was no significant difference in osteoblast cell spreading between modified anodized surfaces A and B. In addition, anodized surfaces A and B showed significantly higher ALP activity and Alizarin red staining than both SLA and machined surfaces ($p < 0.05$). For human gingival fibroblast cells, the modified anodized surfaces exhibited significantly higher cell adherence ($p < 0.05$), followed by the machined surface, with SLA surfaces showing the least adherence.

## Conclusions

The modified anodized surfaces showed superior biocompatibility, enhancing attachment of both osteoblast and fibroblast cells, whereas the SLA surfaces exhibited the least fibroblast cell adherence.

## Introduction

Dental implants are widely recognized as a reliable, long-term solution for tooth replacement, with their success largely dependent on effective osseointegration and soft tissue integration [1–4]. Over recent years, significant advancements in implant surface modification have emerged, aiming to accelerate osseointegration for quicker loading times and to reduce the risk of peri-implantitis by inhibiting biofilm development and attachment [5–7]. Common techniques for creating microtopography on implant surfaces include sandblasting, acid etching, plasma spraying, grit blasting, hydroxyapatite coating, and anodization [8,9]. Research indicates that an optimal implant surface roughness, typically ranging between 1–1.5 µm, promotes the osteoblast cell response, thereby enhances osseointegration [5,10–12]. Despite these advancements, the ongoing challenge remains to identify or develop a surface treatment method that maximizes osteoconductive properties while simultaneously preventing bacterial invasion. Anodization is one such surface modification technique showing promise, as it supports both osseointegration and soft tissue integration through favorable cellular responses [13,14]. However, the impact of anodization on soft tissue behavior, particularly at varying voltages, demands further exploration. Analyzing osteoblast and fibroblast cell adhesion on titanium surface anodized at different voltages can provide valuable insights for selecting the most appropriate implants for specific clinical conditions, ultimately guiding dental implant choices more effectively.

Anodization involves electrochemically treating the implant surface, often following a sandblast large grit and acids etch (SLA) treatment, to create an oxide layer [15]. This oxide layer acts as a corrosion barrier with high surface energy [16]. The thickness of the oxide layer is controlled by the anodizing voltage, which also affects the surface color due to wavelength reflection [17–19]. Recent studies have demonstrated that anodization effectively enhances osteoblast adhesion and proliferation, alkaline phosphatase activity, and calcium deposits, thereby improving osseointegration [13,14,20]. For instance, Traini et al. indicated that anodized implant surface with a titanium oxide ($TiO_2$) coating exhibit lower surface tension and a hydrophilic

surface, which fosters improved blood clot formation [21]. This hydrophilic quality is crucial as it facilitates osteoblast cell attachment to the implant surface, enhancing the osseointegration process necessary for immediate implant loading and long-term success [22].

Maintaining peri-implant soft tissue health is critical for the long-term success of dental implants, as peri-implant infections are the primary cause of implant failure, leading to bone loss [23,24]. Hemidesmosome attachment plays a vital role in limiting bacterial infiltration, thus reducing the risk of infection [25–28]. Anodization enhances both osseointegration and soft tissue attachment through favorable responses from osteoblasts and fibroblasts. *In vitro* studies have demonstrated better fibroblast cell adherence and reduced biofilm formation on anodized surfaces [29,30]. While SLA treatments improve cellular adhesion, they also inadvertently enhance bacterial adherence, which can result in increased inflammation and bone loss compared to polished surfaces [31–33]. In contrast, anodized surfaces exhibit significantly decreased bacterial adherence, presenting a promising alternative for implant surfaces [30,34,35]. Advancements in anodized implant surfaces have shown improved interactions with osteoblasts and gingival fibroblasts. However, there is a lack of comprehensive studies examining cellular responses across different implant surfaces at varying anodization voltages. Our objective is to investigate the effects of different anodization voltages on titanium surfaces, alongside machined and SLA surfaces, focusing on osteoblast and fibroblast cell responses, surface characteristics, and hydrophilicity.

## Materials and methods

The study size was calculated from the pilot study. Twenty type IV titanium discs, each measuring 9 mm in diameter and 3 mm in thickness, were utilized according to ISO 5832–2:2018 standards [36]. The discs were divided into four groups, with each group receiving different surface treatment. The control group was machined titanium surface. The first surface modification is sandblasted with large grit and acid etch (SLA). The titanium discs were sandblasted with aluminum oxide ($Al_2O_3$), followed by acid etching using a mixture of 60% sulfuric acids and 30% hydrochloric acid. Two distinct anodization processes were applied to the surfaces following sandblasting and acid etching. The first one were anodized in a 5% sulfuric solution using a voltage range of 20-30V DC at 0.1 A to get anodized A (blue) surface. The second one were anodized in a 5% sulfuric solution using a voltage range of 60-70V DC at 0.1 A to get anodized B (yellow) surface. After surfaces modifications were completed, all samples underwent a sterilization process by cleansing with alcohol, deionized water, and phosphate buffer saline in preparation for subsequent analysis.

### Surface characterization

The surface roughness of the titanium discs was measured using a surface profilometer (Alicona, infinitefocus, Austria) at x10 magnification. Three different locations on each sample were scanned and measured to obtain representative data. The surface morphology of the titanium discs were assessed using scanning electron microscopy (SEM)(JSM 5410LV, JEOL, Japan) at x5000 magnification. The three-dimensional profile images of different titanium surfaces were measured using a surface profilometer (Alicona, infinitefocus, Austria) at x50 magnification.

The surface wettability was determined using the sessile drop method by measuring the water contact angle [37]. The 10 µL sessile droplet of deionized water was deposited vertically on the surface using a micropipette, ensuring no physical contact with the surface. The contact angle was measured 5 times for each samples to obtain consistent data. The results were reported as mean ± standard deviation.

### Behavior of osteoblasts and fibroblasts on titanium discs

Human osteoblast cells were obtained from bone fragments during dental implant procedures. Written informed consent was obtained from volunteers aged 20−50 years, with approval from the ethics board of Faculty of Dentistry, Chulalongkorn University (HREC-DCU 2022−098). Bone fracments were harvested between 16 December 2022 and 29 May 2023 and cultured in high glucose-Dulbecco modified eagle medium (DMEM Gibco BRL, Ca, USA) containing 10% fetal bovine

 

serum, 2 mM of L-glutamine, penicillin (100 U/ml), streptomycin (100 mg/ml, Gibco) and amphotericin B (5 mg/ml). Induction of differentiation to osteoblasts was promoted by culturing with 10 nM dexamethasone, 25 µg/ml L-ascorbic acid, and 10 mM β-glycerophosphate. Cells from the third to the sixth passages were use for experiment after characterization.

Human gingival fibroblast cells were obtained from the molar teeth of 3 patients, aged between 20–30 years, extracted for orthodontic reasons at the Department of Surgery, the Faculty of Dentistry, Chulalongkorn University (HREC-DCU 2024–011) between 6 May 2024 and 20 August 2024. Gingival fibroblasts were scraped from the middle third of the root and cultured for 3–4 weeks in DMEM containing 10% fetal bovine serum, 2 mM of L-glutamine, penicillin (100 U/ml), streptomycin (100 mg/ml) (Gibco) and amphotericin B (5 mg/ml). Cells from the third to the fifth passages were used in the study.

All tissues used to isolate osteoblasts and fibroblasts were obtained from healthy patients. The patients had no history of systemic disease, smoking, or medication that could affect bone or soft tissue healing. Cells were seeded on polystyrene surface and all titanium discs in 48 well plates at a density of $2 \times 10^4$ cells/surface. The cultures were incubated for 24 hours at 37 °C. The proliferation rate of osteoblasts and fibroblasts was evaluated using the MTT (methylthiazolytetrazolium) assay. Measurements were taken at 24 hours, 3 days, and 7 days post-seeding. At each time point, the culture media was measured at an absorbance of 570 nm using a microplate reader (Elx800; Biotek, Winooski, VT).

### Immunofluorescent staining

Human osteoblast cells were seeded at $2 \times 10^4$ cells per well in 24-well plates for 1 hour and 3 hours. Human gingival fibroblast cells were also seeded at the same density and observed after 3 hours. Cells were rinsed twice with 0.1M PBS, fixed in 4% paraformaldehyde and permeabilized in 0.1% Triton X 100 for 10 minutes. Cells were incubated overnight with primary antibody vinculin (1:100; Abcam, Grand Island, NY) to stain focal adhesion points. Rhodamine-conjugated phalloidin (1:200; Molecular Probes-Life Technologies, Grand Island, NY) was used to stain actin filaments. Nuclei were stained with 300 nM 4′,6-diamidino-2-phenylindole, dihydrochloride (DAPI) for 20 min. A glass coverslip was mounted on the titanium surface containing cells using an antifade kit (Vectashield, Vector Laboratories, Burlingame, CA). Fluorescent images were analyzed using a fluorescent microscope (Axiovert 40CFL, Carl Zeiss, Gottingen, Germany). Cell adhesion was quantified by counting the number of cells in 3 fields of view per sample, across three discs for each titanium surface.

### Alkaline phosphatase (ALP) activity

Osteogenically differentiated human osteoblast cells were seeded on all titanium discs in 48-well plates at the density of $3x10^4$ cells per well. ALP activity was measured at two time points: after 3 and after 7 days. The cells were incubated with 300 µL of BCIP/NBT (5-Bromo-4-chloro-3-indolyl phosphate/Nitro blue tetrazolium) substrate solution (Sigma Life Science, USA) for 30 minutes at 37°C. For quantification, the stained cells were rinsed with 200 µl of 2M KOH to stop the reaction, followed by 300 µl of DMSO to dissolve the intracellular NBT-formazan crystal [38]. The absorbance was measured using a microplate reader at 620 nm (Elx800; Biotek, Winooski, VT) [39].

### Alizarin red S staining

Assessment of mineralization was conducted at two time points: after 14 days and after 21 days. Cells were fixed with cold methanol for 10 minutes. Fixed cells were stained with 1% Alizarin Red S solution (Sigma Life Science, USA) for 5 minutes at room temperature to detect calcium deposits. The quantification was performed by dissolved stained sample in 10% of cetylpyridinium chloride and measured for absorbance at 570 nm.

### Statistical analysis

Statistical analysis was performed using Prism (GraphPad, La Jolla, CA, USA). Shapiro-Wilk's test was used to assess the normality of the data and Levene's test was used to determine the homogeneity of variance. Analysis of Variance

(ANOVA) with Tukey post hoc multiple comparisons were conducted to determine the significance of the results. The data presented for all groups are expressed as standard deviation (SD). Significance levels were categorized as follows: * $p < 0.05$, ** $p < 0.01$, *** $p < 0.001$, **** $p < 0.0001$

## Result

### Surface characteristic analysis

The surface roughness (Ra) and root mean square surface roughness (Sa) values for different titanium surface treatments are presented in the Table 1. Statistical analysis indicated that the machined surface (control) had the lowest Ra (0.33 μm) and Sa (0.38 μm) values, which were significantly different from the treated surfaces. Sandblast and acid-etches surface exhibited increased roughness compared to the machined surface, contributing to a more textured appearance. Anodized surfaces (A and B) showed comparable roughness values to each other, with increased roughness compared to the machined surface. However, anodized A displayed slightly higher mean Ra (1.19 μm) and Sa (1.31 μm) values compared to anodized B (1.15 and 1.29 μm) (Fig 1).

**Table 1. Surface roughness parameter measurement in Arithmetric Mean Roughness (Ra) and Root mean Square Roughness (Sa), different small letters (alphabet) indicated a significant difference between material types.**

| Type of surface modification | Ra (mean SD) | Sa (mean SD) |
|---|---|---|
| Machined surface (Control) | 0.33 μm (0.01)[a] | 0.38 μm (0.01)[a] |
| Sandblasted and acid-etched | 1.08 μm (0.01)[b] | 1.22 μm (0.03)[b] |
| Anodized A | 1.19 μm (0.14)[b] | 1.31 μm (0.24)[b] |
| Anodized B | 1.15 μm (0.06)[b] | 1.29 μm (0.24)[b] |

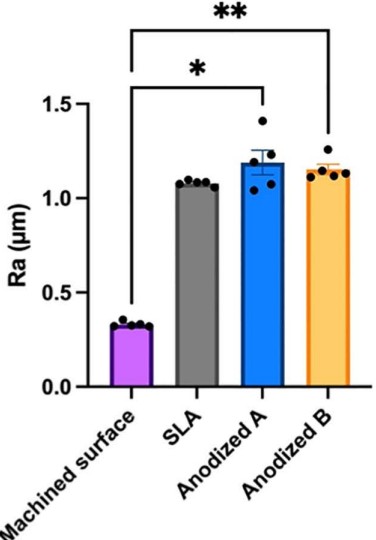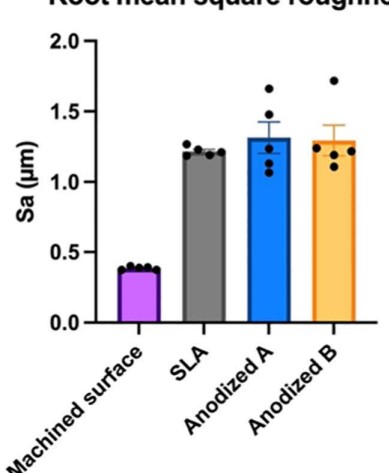

**Fig 1. The surface roughness (Ra) and root mean square surface roughness (Sa) values of different titanium surface treatments.**

The SEM images (Fig 2) revealed that machined surface had relatively uniform, smooth texture characterized by distinct machining lines. SLA surface showed particularly irregular and textured surface morphology. Both anodized surfaces (A and B) exhibited similar rough and pore-filled textures. However, the anodized A surface had deeper valleys, which appeared as pink region in Fig 2.

The surface wettability of different titanium surface treatments was assessed through contact angle measurements, as depicted in Fig 3. Results demonstrated statistically significant differeces in contact angles among all surface treatments ($p < 0.05$). The machined surface showed the highest contact angle of 65.9 degrees, indicating the lowest wettability. The

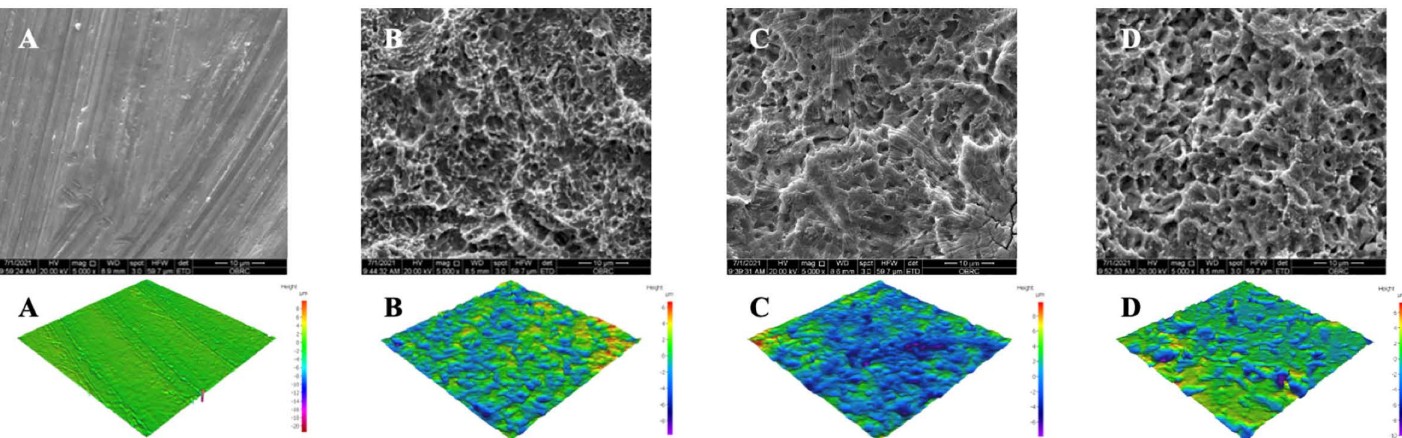

**Fig 2. The SEM image of surface topographic of different titanium surfaces (magnification X5000) and Three-dimensional profile images of different titanium surfaces topographic at X50 magnification, Machined surface (A), Sandblasted and acid-etched surface (B), Anodized A surface (C), Anodized B surface (D). Scale bars = 10 µm.**

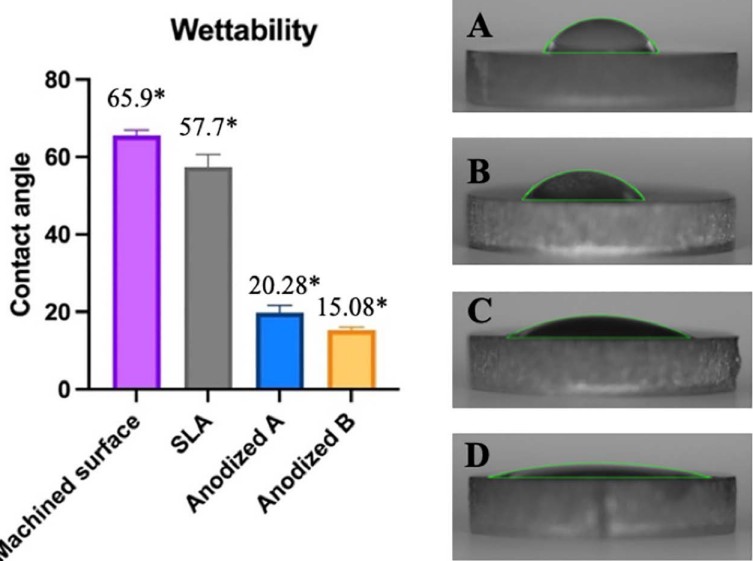

**Fig 3. The average measurement of the contact angle on various titanium surfaces.** Machined surface **(A)**, Sandblasted and acid-etched surface **(B)**, Anodized A surface **(C)**, Anodized B surface **(D)**, Signification level at $p < 0.05$ (*).

SLA-treated surface exhibited a moderately lower contact angle at 57.7 degrees in comparison. The anodized A surface had a notably lower contact angle of 20.28 degrees, while the anodized B surface recorded the lowest contact angle of 15.08 degrees, reflecting the highest level of wettability among the samples.

## Behavior of osteoblastic and fibroblast cells on titanium surface analysis

### Cell cytotoxicity and proliferation

It is observed that both the anodized A and B surfaces showed an increasing proliferation rate when compared to other titanium surfaces (Fig 4). On day 1, the proliferation of the osteoblast cells on the anodized A titanium surface was statistically significantly higher than on all other titanium surfaces ($p < 0.05$). By day 3, the proliferation rate of both anodized A and B surfaces were statistically significantly higher than those of the machine surface and the sandblasted and acid-etched surface ($p < 0.05$). However, no statistically significant difference was found between the anodized A and the anodized B groups ($p < 0.05$). By day 7, there were no significant differences in cell proliferation among test groups, as initial advantages of the anodized A surface diminished overtime. Across time point, significant proliferation changes were found in machined surface titanium between every tested interval (day1 and 3, day 3 and 7, day 1 and 7) The SLA surface exhibited significant differences in proliferation between days 3 and 7 and days 1 and 7 only. Anodized A demonstrated significant changes solely between days 1 and 7, while Anodized B showed significant differences proliferation across all intervals tested.

### Immunofluorescent staining

At the 1-hour mark (Fig 5), both anodized A and anodized B surfaces recruited a higher number of osteoblast cells compared to machined and SLA surfaces. Osteoblasts on these anodized surfaces displayed larger sizes, and the focal adhesion molecules (indicated by green fluorescence) were more abundant than those on machined and SLA surfaces. Co-localization of actin and vinculin at the cell borders, appearing orange, was the most prominent on anodized B surface. By the 3-hour time point, osteoblast cells had extended their cytoskeletons and transitioned from a round morphology to polygonal shapes. The highest number of cells were found on anodized A and B surfaces, followed by the SLA and machined surfaces. Co-localization of actin and vinculin was again most abundant on the anodized B surface.

From Fig 6, a statistically significant difference in the number of gingival fibroblast cells was observed between the anodized titanium A, B, and the machined surfaces compared to the SLA surface. The SLA surfaces contained the lowest number of fibroblast cells. Both the polystyrene coverslip and machined surface demonstrated increased cytoskeletal

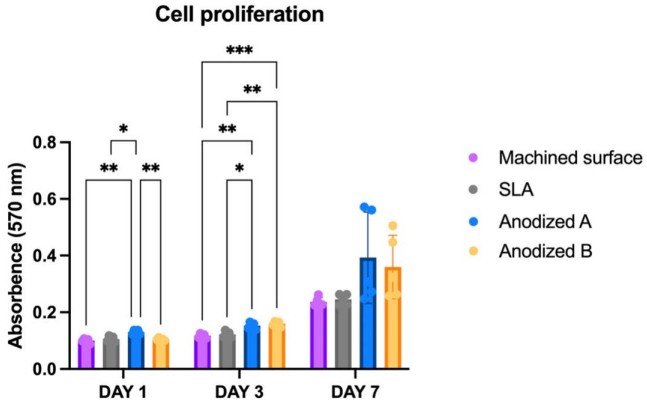

**Fig 4. Osteoblast cell viability and proliferation assay after 1, 3, and 7 days.** Signification level at $p < 0.05$ (*), $p < 0.001$ (**), $p < 0.005$ (***).

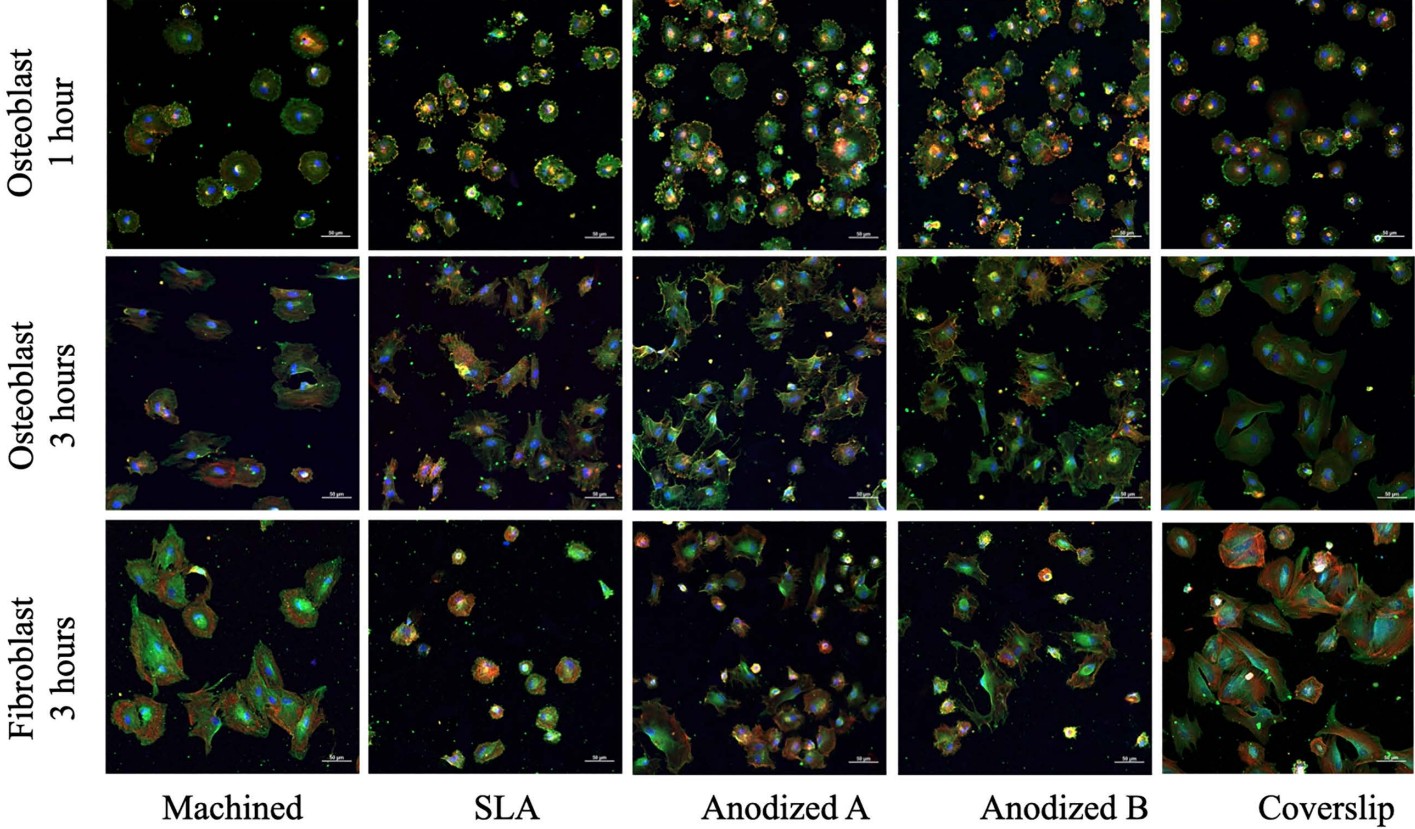

**Fig 5. Immunofluorescent staining of osteoblast and gingival fibroblast cell for 1 and 3 hours.** (magnification X20), The focal adhesion molecule of the cells was identified by vinculin staining (green color). The actin filaments in the cells were highlighted by the phalloidin staining (red color). The nucleus of the cells was visible through the use of DAPI (blue color). Scale bars = 50 μm.

structure and fibroblast cell sizes. Although the anodized A and B surfaces had smaller cell sizes compared to the machined surface, the cytoskeletal distribution and polygonal form were still visible, in contrast to the circular shape and lack of cytoskeletal distribution observed on SLA surfaces.

## Alkaline phosphatase

On day 3, there was no statistically significant difference in the alkaline phosphatase levels observed among various surfaces (Fig 7), although the anodized B surface exhibited the highest level among those tested. By day 7, alkaline phosphatase levels on both anodized A and B surfaces were statistically significantly higher than those measured on the machined surface (control). There was no statistically significant difference in alkaline phosphatase levels between the anodized A, anodized B, and the sandblasted/acid-etched surfaces. When comparing ALP activity over time, statistically significant differences were found at both days 3 and 7 for all titanium surface types.

## Alizarin red

At day 14, alizarin red staining did not differ significantly between groups, though the anodized B showed a slightly increase (Fig 7). By day 21, anodized B exhibited the highest alizarin red levels, indicating greater mineralization compared to other surfaces. Alizarin red levels on both anodized A and B surfaces were statistically significantly

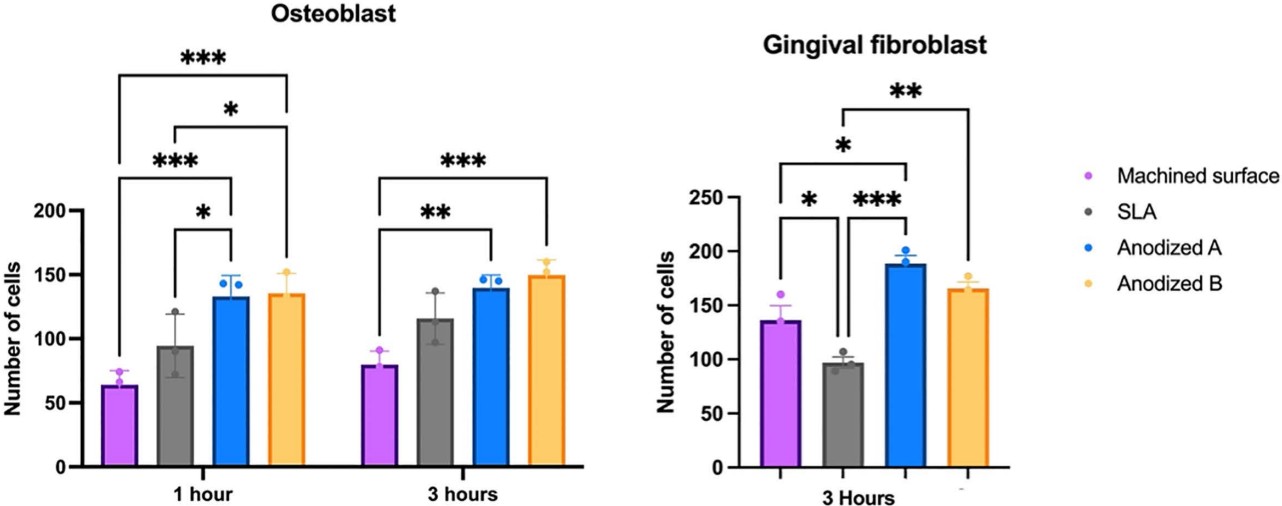

**Fig 6. Cell amount of osteoblast and fibroblast cells on each type of titanium surface at 1 hour and 3 hours.** Signification level at $p < 0.05$ (*), $p < 0.001$ (**), $p < 0.005$ (***).

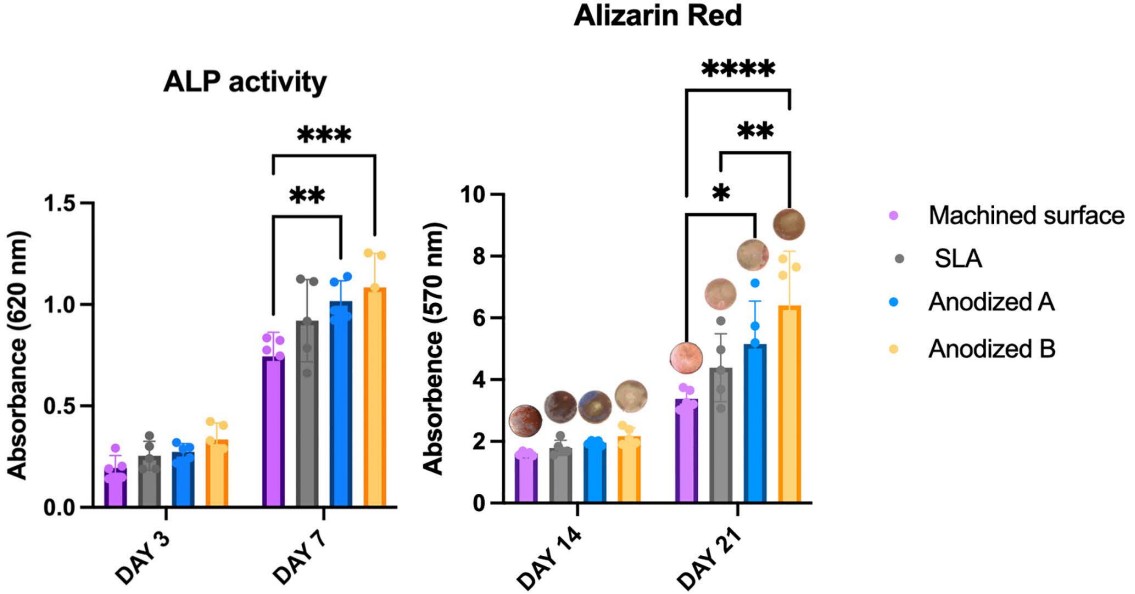

**Fig 7. Alkaline phosphatase activity after 3, and 7 days (left), Alizarin Red test after 14, and 21 days (right), Signification level at $p < 0.05$ (*), $p < 0.001$ (**), $p < 0.005$ (***), $p < 0.0001$ (****).**

higher than those on the machined surface (control). Additionally, the anodized B surface showed a statistically significantly higher alizarin red level compared to SLA surfaces but there was no significant difference between anodized A and B.

In time-dependent comparison, Alizarin Red staining demonstrated statistically significant differences at both days 14 and 21 among all examined titanium surface categories.

## Discussion

Grade IV titanium was chosen for its higher purity, resulting in fewer allergic reactions and better corrosion resistance than Grade V due to the absence of aluminum and vanadium, which supports long-term biocompatibility [40,41]. Previous implant studies have also used Grade IV, making it suitable choice for this experiment [13,42,43]. The anodization process used for anodized A and B surfaces followed sandblasting and acid etching, resulting in a significant increase in surface roughness. This modification created a more pronounced valley and peak topography compared to SLA surfaces, as demonstrated in Fig 2. The roughness of the anodized A and B surfaces, ranging from 1.12 to 1.19 µm, aligns with previous findings that highlight this range as optimal for promoting osteoblast responses [12]. Research indicates that higher anodizing voltages produce rougher surfaces, characterized by numerous and larger micropores, increased crystallinity and a thicker oxide layer. These features collectively enhance osseointegration, as they provide a more suitable environment for bone cell attachment and growth [44–46]. Additionally, the anodized surfaces exhibited the lowest contact angle among different surface treatments tested, indicating superior wettability. The decreased contact angle is directly associated with increased hydrophilicity. Enhanced wettability facilitates better protein adsorption and cell adhesion, thereby promoting cell proliferation and differentiation [47].

This study employed human osteoblast and gingival fibroblast cells to closely replicate realistic cell activities. Previous studies consistently indicate a superior cell response on anodized titanium surfaces compared to machined surfaces [46,48,49]. However, the optimal voltage for promoting osteoblast adhesion, proliferation, and mineralization remains a point of contention in the literature. Some studies have reported that higher voltages yield better cell responses [50,51], whereas others have found no significant differences across various voltages [52,53]. Ercan et al. noted a peak in osteoblast density at 15 volts on day 5 [51]. In this study, we selected lower voltages (20–30 volts) for anodized A and higher voltages (60–70 volts) for anodized B to assess cell proliferation at day 1, 3, 7. Two anodization voltage ranges (20–30 V and 60–70 V) were selected to represent the typical lower and upper limits for forming titanium nanotubular. These ranges, commonly used in research and compatible with available equipment, have also been employed in commercial implant systems (Nobel Biocare and MegaGen) aligning with reported cellular responses [51,54]. The results demonstrated that both anodized A and B surfaces exhibited higher cell growth and cytocompatibility for osteogenesis compared to other titanium surfaces over a 7-day period. These findings align with several studies that have reported enhanced proliferation rates on anodized surfaces compared to machined surface [51,52]. At day 7, cells demonstrated greater adhesion and more extensive spreading on the titanium surface compared to days 1 and 3. Such enhanced attachment and morphology of osteoblasts are favorable for improving implant stability during the healing phase. This finding further supports the clinical concept of immediate loading implants, where rapid and stable cell adhesion to the implant surface is essential for early osseointegration. However, there was no significant difference in osteoblast cells response between the two anodized surfaces. No significant differences in cell adhesion, proliferation, or differentiation were observed between the two anodization voltages for most assays, except for the MTT assay on day 1. This may be explained by a plateau effect, where nanotube morphology and surface chemistry within the tested range were sufficiently similar to elicit comparable cellular responses. It appears that osteoblasts and fibroblasts respond primarily to the presence of nanotubular structures rather than subtle variations in tube diameter. These findings suggest that the general nanotubular architecture is more critical for biological outcomes than precise anodization voltage.

The initial reaction at the tissue-implant interface is cell adhesion. Cell expansion and adherence were detected through the interaction between vinculin (green fluorescence) and actin (red fluorescence), which appeared as an orange color in the immunofluorescent images (Fig 5). The force generated by the expanded actin stress fibers, resulting from the partial coupling of vinculin and talin to actin binding, is known as the "actomyosin force machinery" [55]. Vinculin plays an important role in regulate cell adhesion by connecting with the actin cytoskeleton and stabilizing focal adhesions. Differentiating osteoblasts are often characterized by prominent stress fiber bundles along their borders [56,57]. These findings suggest that vinculin is a key indicator of cellular response and interaction with titanium surfaces over time. Our results

correspond with those of previous studies [50,58], showing that the bright orange and green fluorescence (indicating vin-culin) on anodized A and B surfaces signifies enhanced cell adherence, extension, and differentiation. Immunofluorescent images demonstrated a superior cell response on the anodized titanium surfaces, which was consistent with SEM obser-vations indicating improved osteoblast cell adhesion on anodized surfaces over time (1–3 hours). The hydrophilic nature of the anodized surfaces (characterized by low contact angles) facilitated better osteoblast proliferation, differentiation, and adhesion as evidenced by the presence of filopodia protrusions and microvilli extension.

Normally, fibroblast cells showed good adherence to the machine surface compared to the rough surface (SLA surface) [59,60]. However, a lot of research suggests a microrough surface to promote gingival fibroblast cell attachment [28,61,62]. In this study, the anodized A and B surfaces demonstrated superior cell attachment and spreading compared to the SLA surfaces. Notably, the anodized A surface exhibited the highest fibroblast cell attachment, followed by anodized B, machine surfaces, and the least cell attachment was SLA surfaces respectively. The modified anodized titanium surface (sandblasted before anodization) employed in this work showed results consistent with those of Xu et al., indicating that anodized titanium surfaces improve fibroblast cell responsiveness [63]. Furthermore, several studies have confirmed that fibroblast cells attach more effectively to anodized surfaces in comparison to the control surfaces [29,53,64,65]. Gulati et al. also showed that fibroblast cells exhibit longer filopodia spread and improved cell alignment on the anodized surfaces treated at higher volt-ages. Additionally, these studies noted and increase in integrinβ1 expression, a receptor that mediates cellular binding onto the implant surface [53]. Figs 4 and 5 of this study further corroborate these findings through extensive cell spreading and organized filopodia structures observed via immunofluorescent imaging. The enhanced fibroblast cell response on the anod-ized surfaces, illustrated by the growing filopodia, aligns with findings from prior studies. Specifically, vinculin-actin interaction observed in fibroblast cells appeared as orange color in Fig 5, which was most abundant at the borders of the anodized A surface, indicating the highest fibroblast cell adhesion. Vinculin, a crucial component of focal adhesions and cell mechanosen-sory mechanisms, plays a key role in mediating integrin functions that influence cell motility and behavior [66]. This detailed cellular interaction is critical for the formation of a strong extracellular matrix, thereby enhancing tissue integration and healing. The anodized titanium surface facilitates the gingival fibroblasts adhesion, deposition, and proliferation, which forms a robust bond potentially capable of preventing microbial invasion [6,53,67]. It is noteworthy that human gingival fibroblast response, as observed through immunofluorescence, is superior on the anodized surface compared to the SLA surface, a finding that aligns with the results of Lima et al [29]. The reason is the anodized surfaces had porosities, the $TiO_2$ oxide layer, and the highest wettability, which is favorable for fibroblast deposition, adhesion, and proliferation [29,68].

ALP activity was used to assess initial osteoblast differentiation, while alizarin red staining was employed to evaluate mineralized nodule formation on titanium surfaces. In this study, ALP activity and alizarin red staining were highest on anodized B surfaces, likely influenced by the applied voltage and the thickness of the oxide layer. The hydrophilicity of the anodized surfaces and the presence of the Ti-OH groups on the surface layer likely promoted this enhanced response [47]. When $Ca^{2+}$ ions combine with $OH^-$ groups, the surface gains a positive charge and reacts with phosphate groups to form amorphous calcium phosphate [69]. This observation is supported by Erdogan et al, who found that anodization at the higher voltages resulted in the best alizarin red measurements at 2 weeks, although no significant differences were noted between different anodized voltages [52]. Consequently, increasing anodization voltage appears to enhance oxide layer thickness, improving surface wettability, promoting cell proliferation, spreading, ALP activation, and calcium deposi-tion. In this experiment, although anodized B surfaces exhibited lower contact angles compared to anodized A surfaces, no statistically significant differences were found between ALP activity and alizarin red staining of the anodized A and B surfaces ($p < 0.05$). Fig 3, 6 and 7 indicate that there is no notable difference in bone formation between machined titanium and SLA surfaces. Typically, fibrin from the blood clots helps osteoblasts adhere under normal physiological conditions. However, because our experiment did not included fibrin, osteoblast attachment to the SLA surface was reduced. On the other hand, the anodized surface's higher hydrophilicity enhanced cell adhesion, as demonstrated by the immunofluores-cent images leading to increase bone formation in line with previous studies [21].

Anodized titanium surfaces show enhanced biocompatibility due to a combination of physicochemical and mechanical properties which enhance surface wettability and promote cell spreading and cell proliferation. From the data present in this study, human osteoblast cells and fibroblast cells respond better on anodized surfaces than on SLA and machined. However, the mechanism of human osteoblast cell and fibroblast cell behavior on the anodized titanium at the molecular level needs further investigation.

## Conclusion

The study found that modified anodized titanium surfaces substantially improve osteoblast and fibroblast attachment, proliferation, and differentiation when compared to machined or SLA surfaces. Enhanced surface roughness, porosity and hydrophilicity were identified as key contributors to superior cell responses and stronger tissue integration. These attributes render modified anodized surfaces particularly suitable for applications requiring robust bone and soft tissue integration, such as dental implants. It remains challenging to distinguish the individual effects of surface roughness and hydrophilicity due to the substrates employed; thus, the enhanced cellular response should be regards as a consequence of combined surface features. Notably, high-voltage anodization promotes osteoblast adhesion, whereas low-voltage anodization is more conducive to fibroblast adhesion. Future studies is warrented to optimizie anodization parameters for further improvement in cell responses and to investigate the broader clinical implications of these findings.

## Supporting information

**S1 Data. Cck.**
(XLSX)

**S2 Data. Data all manuscript.**
(XLSX)

## Acknowledgments

This study was supported by the Prosthodontic Department, Faculty of Dentistry, Chulalongkorn University. Center of Excellence for Dental Stem Cell Biology, Chulalongkorn University. Oral Biology Research Center, Chulalongkorn University. Dental Material R&D Center, Chulalongkorn University. We thank Dr. Prasit Pavasant for technical assistance.

## Author contributions

**Formal analysis:** Wareeratn Chengprapakorn, Kamonmard Moungthong, Daneeya Na Nan.

**Investigation:** Kamonmard Moungthong.

**Methodology:** Pravej Serichetaphongse.

**Software:** Kamonmard Moungthong, Daneeya Na Nan.

**Supervision:** Wareeratn Chengprapakorn, Pravej Serichetaphongse.

**Validation:** Daneeya Na Nan.

**Writing – original draft:** Kamonmard Moungthong.

**Writing – review & editing:** Wareeratn Chengprapakorn, Pravej Serichetaphongse.

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
