## [Decision Letter · Decision Letter 0]

27 Aug 2025

Dear Dr. Chengprapakorn,

We look forward to receiving your revised manuscript.

Kind regards,

Tapash Ranjan Rautray

Academic Editor

PLOS ONE

Journal Requirements:

2. Please include a separate caption for each figure in your manuscript.

Additional Editor Comments:

Dear Authors,

Thank you for submitting your manuscript to PLOS One. After thorough evaluation by Reviewers, it was found that your study addresses an important topic relating to the influence of titanium surface anodization on osteoblast and fibroblast behavior. The reviewers acknowledge the relevance of your work. However, both reviewers have raised substantive concerns that must be addressed before the manuscript can be considered for publication.

Reviewers' comments:

Reviewer's Responses to Questions

**Comments to the Author**

1. Is the manuscript technically sound, and do the data support the conclusions?

Reviewer #1: Partly

Reviewer #2: Partly

2. Has the statistical analysis been performed appropriately and rigorously?

Reviewer #1: Yes

Reviewer #2: Yes

3. Have the authors made all data underlying the findings in their manuscript fully available?

Reviewer #1: No

Reviewer #2: Yes

4. Is the manuscript presented in an intelligible fashion and written in standard English?

Reviewer #1: Yes

Reviewer #2: Yes

Reviewer #1: The study by the authors addresses a pertinent, yet unaddressed, question related to the effect of different anodizations of the Ti surface on the cellular response. Using an in vitro setup, the authors demonstrate that anodization, irrespective of its associated technique, improves cell viability, adhesion, and the regenerative response of osteoblasts and fibroblasts in comparison to a machined control. This work is of interest and may motivate future in vivo studies to probe the effects of different anodization protocols on osseointegration prior to any translation to patients.

That being said, the study has several concerns that the authors are invited to address.

1. The biological effect of titanium surface anodization in comparison to a machined titanium surface has been extensively studied in vitro and in vivo. In line with what the authors demonstrated, anodization triggers pro-regenerative signalling. Importantly, this results from coupled modulation of the inflammatory response. Inflammatory cells are the first cellular actors to encounter the titanium surface in tissues. However, the authors largely overlook this crucial aspect and instead focus on the regenerative cells.

The reviewer judges that it is critical for the current work to provide complementary experiments employing cells representative of the initial inflammatory phase, such as THP-1 cells. This would substantially aid the mechanistic understanding of the differential responses between the surfaces, beyond the current interpretation based solely on cell adhesion.

Please address by providing additional data.

2. Why did the authors choose Grade IV titanium, which is most frequently utilised in orthopaedics, while Grade V is mostly exclusive to the dental field? Such a choice does not support the translational value of the study. Please explain and consider adding a dedicated section on this point in the Discussion.

3. How do the authors justify the selected anodization voltages (30 and 70 V)? Please clearly explain the reasoning behind this in the Discussion.

4. While the authors showcase the improved regenerative response via ALP and Alizarin Red assays, confirmation at the gene level appears necessary to support the conclusions. Please complement with additional data.

5. Were the tissues used to obtain osteoblasts and fibroblasts all from healthy patients? Please provide further information on the patients from whom tissues were harvested for the fibroblast and osteoblast experiments.

6. Please complement the ALP assay and the Alizarin Red data with associated microscopy images that support the quantitative data.

7. How do the authors interpret the absence of differences in the biological effects between different surface anodization voltages? Please dedicate a section to this in the Discussion.

8. Please clarify whether the variance shown in all graphs is SEM or SD. This information should be incorporated in the Statistics section of the Methods and in each figure caption.

9. Table 1 does not fully and clearly disclose the intergroup statistics for the comparison of the surface characteristics. Instead, by showing a graph (for example, a bar graph), a reader would more easily note these differences. Please address.

10. While the authors describe the statistical comparison between the experimental groups, no time-dependent comparison is provided within each group. This would help highlight the kinetic changes of the different studied parameters. Please complement accordingly.

11. For transparency purposes, please add plots to the bar graphs.

Reviewer #2: The authors tested the effects of anodization on titanium substrates on osteoblast and fibroblast cell behaviors. The surface topography and surface chemistry of titanium discs were modified with sandblasting/acid etching and anodization. Primary human osteoblasts and fibroblasts were cultured and the authors measured cell response.

Specific Comments:

1. The authors state at the end of the Introduction that “Our objective is to investigate the effects of different anodization voltages on titanium surfaces, alongside machined and SLA surfaces, focusing on osteoblast and fibroblast cell responses, surface characteristics, and hydrophilicity” which suggests the manuscript may be focused on investigating anodization engineering conditions. The authors even write “However, the impact of anodization on soft tissue behavior, particularly at varying voltages, demands further exploration” and ”However, there is a lack of comprehensive studies examining cellular responses across different implant surfaces at varying anodization voltages”. Yet only two voltages were tested (20-30 V and 60-70 V). Is this a sufficient range to test to address gaps in the literature? In all the assays performed, the authors found no statistical difference between the two anodization voltages save for the MTT assay on day 1. The authors point out in the Discussion that anodization voltages did have statistically significant effects in some studies, but not here and offer no further discussion.

2. The Conclusion states “The increased surface roughness and porosity, along with enhanced hydrophilicity, contribute to better cell responses and stronger tissue integration”. The effects of surface roughness and hydrophilicity are not clearly demonstrated by the choice of substrates and the presented data. The substrates used in this study are: machined (control), sandblasted and acid etched (SLA), SLA + anodization at 20-30 V, and SLA + anodization at 60-70 V. Anodized surfaces have roughness, so what is the appropriate control surface to test the effects of anodization? Is there a synergistic effect of roughness and hydrophilicity? It is not clear how one can conclude that roughness or hydrophilicity independently have an effect on cellular responses given these choices of substrates.

For the effect of surface roughness, the comparison of the control to SLA is a valid comparison. However, the presented data and statistical analysis do not support the conclusion that roughness has an effect. Figures 3, 6, 7 shows no statistical difference between machined surfaces and SLA, save for cell adhesion for fibroblasts on hour 3.

3. Figure 3 shows results of the MTT assay after 1,3, and 7 days. The error bars are quite consistent across measurements, except on day 7. Any explanation? Was any outlier test performed?

4. Figure 4 shows SEM images of cells. The authors describe morphological changes of cells on different substrates. SEM preparation is harsh for cells (fixation, washes, drying) and morphology may not be representative of in vitro, much less in vivo conditions. How useful is SEM imaging for cell morphology characterization? The authors performed fluorescent imaging, which may be a better dataset to perform morphological analysis. Furthermore, morphology such as shape and size can be quantified. The authors should support their discussion of morphological changes with quantitative image analysis.

Minor comments: Scale bars on SEM and fluorescence images are difficult to read, please enlarge. Suggest labeling sample conditions in the figure instead of just in the figure caption to help the reader.

5. The manuscript mentions a pilot study in the Materials and Methods section. What are the details of this pilot study? These details may would be suitable for Supporting Materials.

**Do you want your identity to be public for this peer review?** For information about this choice, including consent withdrawal, please see our Privacy Policy

Reviewer #1: No

Reviewer #2: No

---

## [Author Response · Author response to Decision Letter 1]

15 Oct 2025

Letter to Reviewer 1

1. The biological effect of titanium surface anodization in comparison to a machined titanium surface has been extensively studied in vitro and in vivo. In line with what the authors demonstrated, anodization triggers pro- regenerative signalling. Importantly, this results from coupled modulation of the inflammatory response. Inflammatory cells are the first cellular actors to encounter the titanium surface in tissues. However, the authors largely overlook this crucial aspect and instead focus on the regenerative cells.

The reviewer judges that it is critical for the current work to provide complementary experiments employing cells representative of the initial inflammatory phase, such as THP-1 cells. This would substantially aid the mechanistic understanding of the differential responses between the surfaces, beyond the current interpretation based solely on cell adhesion.

- We sincerely thank the reviewer for this insightful and valuable comment. We fully agree that inflammatory cells represent the first line of interaction with the titanium surface and play a critical role in orchestrating subsequent regenerative processes. We acknowledge that the present study focused primarily on regenerative cells, and that the absence of experiments involving inflammatory cells, such as THP-1 cells, is a limitation.

Unfortunately, as primary human-derived cells were employed in the present work, the limited availability of these cells precluded us from conducting further experiments with inflammatory cell models at this stage. We will certainly consider incorporating inflammatory cell models in our future studies to provide a more comprehensive mechanistic understanding of the biological responses to anodized versus machined titanium surfaces.

2. Why did the authors choose Grade IV titanium, which is most frequently utilised in orthopaedics, while Grade V is mostly exclusive to the dental field? Such a choice does not support the translational value of the study. Please explain and consider adding a dedicated section on this point in the Discussion.

- The dedicated section has been added to the Discussion at page 16

Grade IV titanium was chosen for its higher purity, resulting in fewer allergic reactions and better corrosion resistance than Grade V due to the absence of aluminum and vanadium, which supports long-term biocompatibility [40, 41]. Previous implant studies have also used Grade IV, making it suitable choice for this experiment[13, 42, 43].

3. How do the authors justify the selected anodization voltages (30 and 70 V)? Please clearly explain the reasoning behind this in the Discussion.

- Justification for the selected voltages was added to the Discussion at page 17

Two anodization voltage ranges (20–30 V and 60–70 V) were selected to represent the typical lower and upper limits for forming titanium nanotubular. These ranges, commonly used in research and compatible with available equipment, have also been emploted in commercial implant systems (Nobel Biocare and MegaGen) aligning with reported cellular responses[51, 54].

4. While the authors showcase the improved regenerative response via ALP and Alizarin Red assays, confirmation at the gene level appears necessary to support the conclusions. Please complement with additional data.

- We sincerely thank the reviewer for this important question. As primary human-derived cells were employed in the present work, the limited availability of these cells precluded us from conducting further experiments with inflammatory cell models at this stage. However, we sincerely appreciate the suggestion and will take it into consideration for future investigations.

5. Were the tissues used to obtain osteoblasts and fibroblasts all from healthy patients? Please provide further information on the patients from whom tissues were harvested for the fibroblast and osteoblast experiments.

- The dedicated section has been added to the Materials and Methods page 6

We sincerely thank the reviewer for this important question. All tissues used to isolate osteoblasts and fibroblasts were obtained from healthy patients undergoing third molar extraction or implant surgery. The donors had no history of systemic disease, smoking, or medication-usage that could affect bone or soft tissue healing. Ethical approval for the use of these tissues was obtained from the ethics board of Chulalongkorn University, and written informed consent was provided by all participants.

6. Please complement the ALP assay and the Alizarin Red data with associated microscopy images that support the quantitative data.

- We have incorporated the reviewer’s recommendation in providing additional data on titanium surfaces, specifically images from the Alizarin Red staining test.

7. How do the authors interpret the absence of differences in the biological effects between different surface anodization voltages? Please dedicate a section to this in the Discussion.

- The recommended section has been added to the Discussion at page 17

This may be explained by a plateau effect, where nanotube morphology and surface chemistry within the tested range were sufficiently similar to elicit comparable cellular responses. It appears that osteoblasts and fibroblasts respond primarily to the presence of nanotubular structures rather than subtle variations in tube diameter. These findings suggest that the general nanotubular architecture is more critical for biological outcomes than the precise anodization voltage.

8. Please clarify whether the variance shown in all graphs is SEM or SD. This information should be incorporated in the Statistics section of the Methods and in each figure caption.

- The information has been added to the Statistics section at page 8

For clarification, the data presented for all groups are expressed as standard deviation (SD).

9. Table 1 does not fully and clearly disclose the intergroup statistics for the comparison of the surface characteristics. Instead, by showing a graph (for example, a bar graph), a reader would more easily note these differences. Please address.

- A bar graph has been included to facilitate clearer interpretation and improve the reader’s understanding of the results.

10. While the authors describe the statistical comparison between the experimental groups, no time-dependent comparison is provided within each group. This would help highlight the kinetic changes of the different studied parameters. Please complement accordingly.

- The additional time-dependent comparison has been added at page 10, 12

We thank the reviewer for the comment. In the revised analysis, our MTT cell proliferation results showed that machined titanium exhibited statistically significant differences between days 1 and 3, days 3 and 7, and days 1 and 7. The SLA surface exhibited significant differences between days 3 and 7, days 1 and 7, with no significant difference between days 1 and 3. Anodized A demonstrated significant differences only between days 1 and 7, with no significance detected between days 1 and 3 or days 3 and 7. In contrast, Anodized B exhibited statistically significant differences across all comparisons (days 1 and 3, days 3 and 7, and days 1 and 7). Furthermore, ALP activity showed statistically significant differences at both days 3 and 7 across all titanium surface types, while Alizarin Red staining demonstrated statistically significant differences at both days 14 and 21 across all titanium surface types.

11. For transparency purposes, please add plots to the bar graphs.

- The recommended bar plot graphs were added to enhance clarity and facilitate easier interpretation of the results.

Letter to Reviewer 2

1. The authors state at the end of the Introduction that “Our objective is to investigate the effects of different anodization voltages on titanium surfaces, alongside machined and SLA surfaces, focusing on osteoblast and fibroblast cell responses, surface characteristics, and hydrophilicity” which suggests the manuscript may be focused on investigating anodization engineering conditions. The authors even write “However, the impact of anodization on soft tissue behavior, particularly at varying voltages, demands further exploration” and ”However, there is a lack of comprehensive studies examining cellular responses across different implant surfaces at varying anodization voltages”. Yet only two voltages were tested (20-30 V and 60-70 V). Is this a sufficient range to test to address gaps in the literature? In all the assays performed, the authors found no statistical difference between the two anodization voltages save for the MTT assay on day 1. The authors point out in the Discussion that anodization voltages did have statistically significant effects in some studies, but not here and offer no further discussion.

- Further discussion has been added to page 17

Two anodization voltage ranges (20–30 V and 60–70 V) were selected to represent the typical lower and upper limits for forming titanium nanotubular. These ranges, commonly used in research and compatible with available equipment, have also been emploted in commercial implant systems (Nobel Biocare and MegaGen) aligning with reported cellular responses[51, 54].

- Further discussion has been added to page 17

No significant differences in cell adhesion, proliferation, or differentiation were observed between the two anodization voltages for most assays, except for the MTT assay on day 1. This may be explained by a plateau effect, where nanotube morphology and surface chemistry within the tested range were sufficiently similar to elicit comparable cellular responses. It appears that osteoblasts and fibroblasts respond primarily to the presence of nanotubular structures rather than subtle variations in tube diameter. These findings suggest that the general nanotubular architecture is more critical for biological outcomes than precise anodization voltage.

2. The Conclusion states “The increased surface roughness and porosity, along with enhanced hydrophilicity, contribute to better cell responses and stronger tissue integration”. The effects of surface roughness and hydrophilicity are not clearly demonstrated by the choice of substrates and the presented data. The substrates used in this study are: machined (control), sandblasted and acid etched (SLA), SLA + anodization at 20-30 V, and SLA + anodization at 60-70 V. Anodized surfaces have roughness, so what is the appropriate control surface to test the effects of anodization? Is there a synergistic effect of roughness and hydrophilicity? It is not clear how one can conclude that roughness or hydrophilicity independently have an effect on cellular responses given these choices of substrates.

For the effect of surface roughness, the comparison of the control to SLA is a valid comparison. However, the presented data and statistical analysis do not support the conclusion that roughness has an effect. Figures 3, 6, 7 shows no statistical difference between machined surfaces and SLA, save for cell adhesion for fibroblasts on hour 3.

- We thank the reviewer for this important methodological point. We agree that with the current substrate set (machined, SLA, SLA + anodization at 20–30 V, SLA + anodization at 60–70 V) it is not possible to attribute observed biological effects uniquely to either surface roughness or hydrophilicity because the anodization treatment alters both parameters simultaneously. Therefore, the present study can only support the conclusion that the combined surface modifications produced by anodization (increased roughness together with increased hydrophilicity) are associated with improved cell responses and mineralization; it cannot demonstrate independent effects of roughness or hydrophilicity. Complementary characterization and analyses should include AFM/profilometry (quantitative roughness metrics), XPS/EDS (surface chemistry), and statistical tests for interaction (e.g., two-way ANOVA) to determine whether effects are additive or synergistic. Because these additional controls were not included in the current manuscript, we have revised the Conclusion and Discussion to make this limitation explicit and to avoid implying independent causation. We now state that anodization-produced changes in roughness and hydrophilicity together correlate with enhanced cell responses, and we have added the above experimental approaches to the Future Work section as planned studies to disentangle independent and synergistic effects.

- The conclusion was changed at page 17

It remains challenging to distinguish the individual effects of surface roughness and hydrophilicity due to the substrates employed; thus, the enhanced cellular response should be regards as a consequence of combined surface features

- the explanation was added at page 17

Figure 3, 6 and 7 indicate that there is no notable difference in bone formation between machined titanium and SLA surfaces. Typically, fibrin from the blood clots helps osteoblasts adhere under normal physiological conditions. However, because our experiment did not included fibrin, osteoblast attachment to the SLA surface was reduced. On the other hand, the anodized surface’s higher hydrophilicity enhanced cell adhesion, as demonstrated by the immunofluorescent images leading to increase bone formation in line with previous studies [21].

3. Figure 3 shows results of the MTT assay after 1,3, and 7 days. The error bars are quite consistent across measurements, except on day 7. Any explanation? Was any outlier test performed?

- We thank the reviewer for this comment. The increased variability observed, particularly on day 7, is most likely attributable to biological differences in cell growth rates among primary human-derived cells from different donors, rather than to experimental error. All data points were carefully examined, and no outliers were identified.

4. Figure 4 shows SEM images of cells. The authors describe morphological changes of cells on different substrates. SEM preparation is harsh for cells (fixation, washes, drying) and morphology may not be representative of in vitro, much less in vivo conditions. How useful is SEM imaging for cell morphology characterization? The authors performed fluorescent imaging, which may be a better dataset to perform morphological analysis. Furthermore, morphology such as shape and size can be quantified. The authors should support their discussion of morphological changes with quantitative image analysis.

Minor comments: Scale bars on SEM and fluorescence images are difficult to read, please enlarge. Suggest labeling

sample conditions in the figure instead of just in the figure caption to help the reader.

- SEM imaging was performed as the initial protocol, while fluorescence imaging was subsequently incorporated to evaluate cell adhesion on the titanium surface. We agree that fluorescence images provide a more robust dataset for morphological analysis and have accordingly followed the reviewer’s suggestion to utilize them. The scale bars and sample conditions have been clearly provided in the revised figures according to the reviewer’s recommendation.

5. The manuscript mentions a pilot study in the Materials and Methods section. What are the details of this pilot study? These details may would be suitable for Supporting Materials.

- We thank the reviewer for raising this important point. The pilot study mentioned in the Materials and Methods section was conducted to optimize the experimental conditions prior to the main study. Specifically, it was performed to validate cell seeding density, culture duration, and staining protocols on the titanium discs to ensure reproducibility and adequate cell attachment for subsequent assays. The relevant details have been previously published online in ThaiJo under the title “Characteristics and osteoblast cell responses of the anodized titanium and sandblasted and acid-etched titanium surfaces.”

---

## [Editor Report · Decision Letter 1]

29 Oct 2025

The impact of anodization modification on titanium interaction with human osteoblasts and fibroblasts (in vitro study)

PONE-D-25-30704R1

Dear Dr. Chengprapakorn,

We’re pleased to inform you that your manuscript has been judged scientifically suitable for publication and will be formally accepted for publication once it meets all outstanding technical requirements.

Kind regards,

Tapash Ranjan Rautray

Academic Editor

PLOS ONE

Additional Editor Comments (optional):

The authors have now complied with the comments of the Reviewers. So, it can be accepted in present form
---

## [Editor Report · Acceptance letter]

PONE-D-25-30704R1

PLOS ONE

Dear Dr. Chengprapakorn,

I'm pleased to inform you that your manuscript has been deemed suitable for publication in PLOS ONE. Congratulations! Your manuscript is now being handed over to our production team.

Kind regards,

on behalf of

Dr. Tapash Ranjan Rautray

Academic Editor

PLOS ONE